# RETRA: Recurrent Transformers for Learning Temporally Contextualized Knowledge Graph Embeddings

Simon Werner[1], Achim Rettinger[1], Lavdim Halilaj[2], and Jürgen Lüttin[2]

[1] Trier University, Trier, Germany
{werners, rettinger}@uni-trier.de
[2] Bosch Research, Renningen, Germany
{lavdim.halilaj, juergen.luettin}@de.bosch.com

**Abstract.** Knowledge graph embeddings (KGE) are vector representations that capture the global distributional semantics of each entity instance and relation type in a static Knowledge Graph (KG). While KGEs have the capability to embed information related to an entity into a single representation, they are not customizable to a specific context. This is fundamentally limiting for many applications, since the latent state of an entity can change depending on the current situation and the entity's history of related observations. Such context-specific roles an entity might play cannot be captured in global KGEs, since it requires to generate an embedding unique for each situation.

This paper proposes a KG modeling template for temporally contextualized observations and introduces the Recurrent Transformer (RETRA), a neural encoder stack with a feedback loop and constrained multi-headed self-attention layers. RETRA enables to transform global KGEs into custom embeddings, given the situation-specific factors of the relation and the subjective history of the entity.

This way, entity embeddings for down-stream Knowledge Graph Tasks (KGT) can be contextualized, like link prediction for location recommendation, event prediction, or driving-scene classification. Our experimental results demonstrate the performance gains standard KGEs can obtain, if they are customized according to the situational context.

**Keywords:** Knowledge Graph Embedding · Contextualized Embeddings · Modeling Temporal Context.

## 1 Motivation

We all play different roles in our lives. In private settings we might act differently than in professional settings. What we represent in a situation depends on contextual factors and there is not a single universally valid representation that captures all roles of a person equally well. In contrast, standard Knowledge Graph Embedding (KGE) methods produce a single vector representation for entity instances and relation types in the corresponding Knowledge Graph

(KG). Each embedding captures the global distributional semantic of the KG in respect to this entity and is optimized for predicting universally valid facts, a Knowledge Graph Task (KGT) known as link prediction. This assumption of universality rarely holds in real-world inference tasks, since the situational context is crucial for making nuanced predictions. When trying to contextualize global KGEs to a situation, two Research Questions (RQ) come to mind:

**RQ1:** How can concrete situations, specifically situational context, be modelled appropriately within knowledge graphs?
**RQ2:** How can static knowledge graph embeddings be transformed into contextualized representations, that capture the specifics of a concrete situations?

In this paper we argue, that a single static KGE per entity and relation is not adequate for many KGTs. Instead, entities and relations need to be put into context by factors specific to the current situation and their subjective history.[3] This requires a different entity embedding for each situation, not just one that attempts to be universally valid. Consequently, there is the need to customize static KGEs to situational and subjective contexts. More precisely, we argue that current models cannot generate relation embeddings that capture the situation-specific relational context (we refer to this limitation as *(Lim1)*) and entity embeddings that contain the subject's history of related observations (we refer to this limitation as *(Lim2)*).

**W.r.t. RQ1:** We propose *temporally contextualized KG facts* (tcKG facts) as a modelling template for situation-specific information in a KG. This adds a temporal sequence of hyper-edges (time-stamped subject-relation-object triples where the relation is $n$-ary in order to capture $n$ contextualizing factors) to an existing static KG (see Sec. 3).
**W.r.t. RQ2:** We contribute the deep learning framework RETRA, which transforms static global entity and relation embeddings into temporally contextualized embeddings, given corresponding tcKG facts. This situation-specific embedding reflects the role an entity plays in a certain context and allows to make situational predictions (see Sec. 4). RETRA uses a novel *recurrent architecture* and a *constrained multi-headed self-attention layer* that imposes the relational structure of temporally contextualized KG facts during training (see Sec. 5).

In order to demonstrate how broadly applicable tcKG and RETRA are we apply and test them in three diverse scenarios, namely location recommendation, event prediction and driving-scene classification. Our empirical results indicates that contextualizing pre-trained KGEs boosts predictive performance in all cases (see Sec. 6).

---

[3] In psychology and neuroscience this distinction might be referred to as semantics vs. episodic memory (see [19]).

## 2    Related Work

Our work attempts to transfer the success of contextualizing word embeddings in Natural Language Processing to Knowledge Graph Embeddings (KGE).

### 2.1    Contextualized Word Embeddings

Word embeddings have been the driving force in Natural Language Processing (NLP) in recent years. Soon after the learning of static embeddings of lexical items became popular their drawbacks became apparent since they conflate all meanings of a word into a single point in vector space. More precisely, static semantic representations suffer from two important limitations: *(Lim1)* ignoring the role of situational context in triggering nuanced meanings; *(Lim2)* due to restricting the scope of meaning to individual entities, it is difficult to capture higher order semantic dependencies, such as compositionality and sequential arrangements between entities. Both limitations were recognized early and addressed by approaches that generate contextualized word representations given surrounding words in a sentence. Before the now dominant transformer approach [20], LSTMs where used to contextualize word embeddings [7]. In the area of KGE the need for contextualizing embeddings has not gotten much attention yet, as we will outline next.

### 2.2    Knowledge Graph Embedding

In recent years KGE has been a very vibrant field in Machine Learning and Semantic Technologies (see [8] for a survey). KGE methods can be roughly characterized by the representation space and the scoring function:

The representation space is traditionally Euclidean $\mathbb{R}^d$, but many different spaces like Complex $\mathbb{C}^d$ (e.g., in [18]) or Hypercomplex $\mathbb{H}^d$ (cmp. [25]) have been used as well. In this work we focus on Euclidean vector spaces only, since they are used for Neural Network embedding models like our RETRA.

The scoring function measures the plausibility of an (unknown) subject-predicate-object triple (referred to as "fact"), given the model parameters. The function produces a scalar score that is obtained by an additive or a multiplicative combination of subject, predicate and object embedding. In this work we focus only on optimizing a given KGE without altering its scoring function.

Standard KGE methods don't take into account temporal information or contextual factors that may influence the plausibility of a fact. In this work we are trying to complement static KGEs without replacing them. To the best of our knowledge there is no existing KGE method that attempts to address both limitations, but they have been addressed individually, as detailed next.

### 2.3    Knowledge Hypergraph Embedding

Approaches to embedd contextualized KG facts is not in the center of current KGE research. However, the use of *n*-ary relations and the modeling of context as a hypergraph has been proposed before the KGE hype. Such approaches

from Statistical Relational Learning were based on graphical models and tensor factorization [16]. A more recent approach extends the current KGE method SimplE [10] to hypergraphs [4] but does not take into account temporal or sequential information. Thus, those approaches address *(Lim1)*, but not *(Lim2)*. In addition, they don't allow to input standard KGE models to transform their embeddings into contextualized KGEs.

### 2.4    Temporal KGE

Embedding temporal dynamics of a knowledge graph and thus tackling *(Lim2)* has seen some attention recently. Basic approaches to temporal KGE, model facts as temporal quadruples. They are optimized for scoring the plausibility of (unkown) facts at a given point in time [11], [3]. A more sophisticated approach is proposed in [14]. It checks the temporal consistency given contextual relations of the subject and object. Besides the inability of those models to model $n$-ary sequential context, we are also taking a different focus by using the temporal dimension to model the history of experiences of a subject.

A more entity-centric perspective is taken in [17] which attempts to model the temporal evolution of entities. This comes close to what we attempt regarding *(Lim2)*, but again, it does not cover $n$-ary relations and is not intended to transform given embeddings into contextualized ones, if provided with a history of subjective experiences. [9] take a relation-specific perspective instead, but still suffer from the same limitations as the above mentioned techniques.

### 2.5    Contextualized KGE

Central to our RETRA-approach is its ability to transform static input embeddings into contextualized KGEs. A similar approach and the same perspective is being adopted in [22]. There, entities and relations are expected to appear in different graph contexts and consequently should change their representation according to the context *(Lim1)*. We agree with this perspective, but argue that temporal evolution is equally important *(Lim2)*. [21] attempt a relation-specific embedding of entities and propose an LSTM-based approach to so. While RETRA is also inspired by Recurrent Neural Networks, we take a more subject-specific perspective. Besides that, temporal and n-ary relations are not considered in [21]. The same limitations apply to [24], but the use of a Transformer is similar to our approach. However, word embeddings are used as inputs to the transformation function instead of pre-trained static KGEs in RETRA.

Summing up, RETRA offers a unique combination that no previous method has attempted: n-ary relations *(Lim1)* and sequential subjective experience *(Lim2)* are exploited to transform static KGE into contextualize ones. RETRA achieves this by two major technical novelties: Modeling KGs with temporally contextualized facts and extending Transformers with a feedback loop and a constrained self-attention layer.

## 3   Modeling Subjective Temporal Context

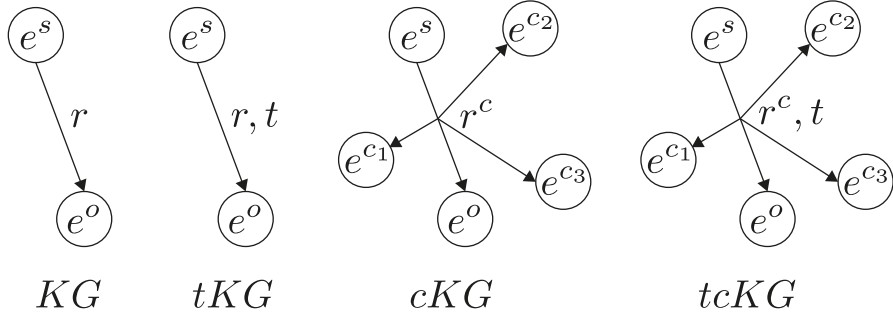

**Fig. 1.** From subject-predicate-object KG triples, to temporal tKG facts which occur at time $t$, to contextualized cKG facts which allow to model influencing factors as an n-ary relation, to tcKG facts which contextualize the KG observation at a certain time. This models the observations of a sequence of relations $r^c$ the subject $e^o$ is involved in.

To address *RQ1* we start from the assumption that static KG facts act as background knowledge but the inference task depends on the situational context and the subject's memory. Consequently, we need to extend triples as follows:

**KG facts** are defined as a triple $(e^s, r, e^o)$ where $e^s, e^o \in \{e^1, ..., e^{n_e}\}$ is from the set of $n_e$ entity instances and $r \in \{r^1, ..., r^{n_r}\}$ from the set of $n_r$ relation types. KG facts constitute subject-predicate-object statements that are assumed as being static and stable background knowledge.

**tKG facts** are quadruples $(e^s, r, e^o, t)$ where $t \in \mathbb{N}$ indicates a point in a sequence when the fact occurred. In many scenarios $t$ is obtained from discretizing timestamps and thus creates a globally ordered set of facts, where $n_t$ is the total number of points in time (cmp. [17])[4]. KG facts without a temporal dimension are considered true for any $t$.

**cKG facts** are $(n+1)$-tuples $(e^s, r^c, e^o, e^{c_1}, ..., e^{c_{n_c}})$ that allow to model context as an $n_c$-ary relation. $e^{c_1}, ..., e^{c_{n_c}}$ are the $n_c$ context entities influencing the relation $r^c$ between subject $e^s$ and object $e^o$. As for KG facts, cKG facts are considered non-dynamic background knowledge given the current situation.

**tcKG facts** are $(n+2)$-tuples $(e^s, r^c, e^o, t, e^{c_1}, ..., e^{c_{n_c}})$ which represent sequentially contextualized KG facts by combining the features of tKGs and cKGs. Intuitively, they capture a specific situation which subject $e^s$ is experiencing at time $t$. $e^c$ are influencing factors towards $e^s$'s relation to object $e^o$.

---

[4] Please note, that temporal KGs have mostly been using $t$ to model the point in time when a fact is being observed. Here, we are taking a slightly different perspective by modeling in which point in time a subject $e^s$ makes an experience in relation to similar experiences it has made at previous points in time.

With tcKG facts as additional building blocks, we can now model task-specific temporally contextualized KGs as temporal hypergraphs (i.e., relations are potentially $n_c$-ary and potentially associated with timestamps $t$). This subjective temporal context is input to RETRA as follows:

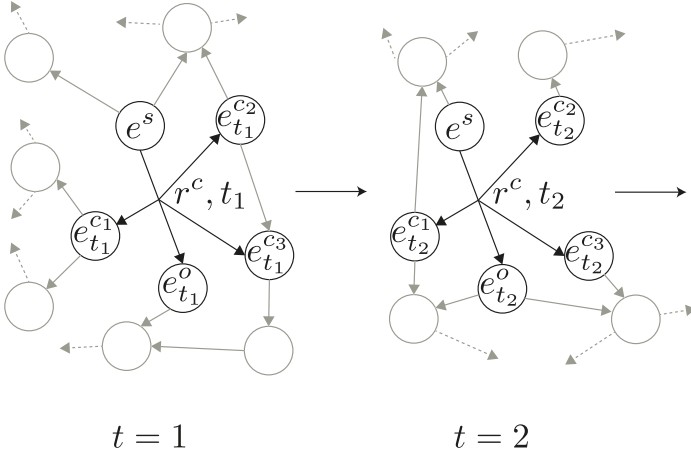

$t = 1$                    $t = 2$

**Fig. 2.** tcKG facts (shown in black) and their relation to static background knowledge (KG and cKG facts, shown in gray) as a temporally unrolled KG.

1. Given an $n_c$-ary relation $r^c$ we first define one entity participating in $r^c$ as the subject $e^s$ whose perspective is represented in respect to an object $e^o$.[5]
2. The contexts $c_1, ..., c_{n_c}$ are given by the remaining entities involved in the $n_c$-ary relation. They define the influencing factors in a concrete situation.
3. If available, the context can be extended by entities deterministically dependent on $e^o$. Non-deterministically dependent context of $e^o$ or $e^c$ or facts that are not specific to a certain point in time $t$ only, are not explicitly modelled (see gray edges and gray nodes in Fig. 2).
4. Finally, the temporal context is modeled by $n_c$ relation instances of $r^c$ that involve $e^s$ as the subject. Sorted by time-stamp $t$, $r^c$ defines the sequential context from the perspective of $e^s$ towards its relation to $e^o$ (see black edges an black nodes in Fig. 2). Note, that the context entities $e^c$ do change in every step, as does $e^o$, thus $e^o_{t_1} \neq e^o_{t_2}$. Consequently, all the facts associated to $e^c$ and $e^o$ do change in every step (gray edges and gray nodes in Fig. 2). Only $e^s$ and the relation-type of $r^c$ stay fixed as defined above.

---

[5] Note, that this is a deliberate modeling choice that is not due to technical limitations. RETRA can model any sets of subjects and objects since transformers allow variable numbers of inputs and can mask any subset during training. We chose this restriction, since this is pragmatically the most common pattern and avoids a cluttered notation.

With the above selection procedure we obtain a sequence of tcKG facts from the KG by filtering for relations $r^c$ with subject $e^s$. Consequently, such a model of dynamic context consists of a sequence of $(e^s, r^c)$-tuples with varying sequence-length $n_t$. In each step $r^c$ has an varying object $e^o$ and is characterized by $n_c$ contextual factors $e^c$.[6]

For illustration purposes, Table 1 shows instantiations of the tcKG modelling pattern according to our three applications domains *location recommendation*, *event prediction* and *driving-scene classification* (see Sec. 6 for details).

| Application | Subject $e^s$ | Relation $r^c$ | Object $e^o$ | Contexts $c_1, ..., c_{n_c}$ |
|---|---|---|---|---|
| Location recommendation | user | checksIn | location | time of day, weather , day of week, location type... |
| Event prediction | source actor | eventType | involved target organizations | target country, source country, sector,... |
| Driving-scene classification | ego vehicle | involvedIn | conflict-type | ego lane, foe road users, foes' lanes, signaling, acceleration, speed,... |

**Table 1.** Illustrating examples of instantiated tcKG patterns for three applications.

## 4    Embedding Subjective Temporal Context

So far, the tcKG modelling pattern provides an explicit representation of dynamic context of a subject and a relation-type as a sequence of sub-graphs (see Fig. 2). The second contribution of this paper, addressing RQ2, is a machine learning method that captures this information in two embeddings, the subjective context $e^s$ and the relational context $r^c$.[7] Once we obtain those embeddings we then can use any embeddings-based KGT scoring functions, e.g., for contextualized link prediction.

One way to capture dynamic context in a single embedding is to represent the history of sequential information in a latent state. As common in Hidden Markov Models or Recurrent Neural Nets, all $t-1$ previous contextualized observation are reduced into one embedding capturing the latent state up to this point. In our model, this memory is captured in the $e^s$ embedding. We thus define the probability $P$ of the contextualized relation representation $r^c$ as being conditioned on $P(r^c|e^s, r, e^o, e^{c_1}, ..., e^{c_{n_c}})$. The subjective context representation $e^s$ depends on $r^c$ but also on the previous experience $e^s_{t-1}$ in similar situations: $P(e^s_t|e^s_{t-1}, r^c_t, e^o_t)$. These conditional dependencies are visualized in Fig. 3.

---

[6] Note, that the arity $n_c$ does not need to be fixed in each step and for each $e^s$. Variable-length context, unknown or missing $e^c$s can be modelled and handled efficiently with RETRA, since transformers can handle variable input lengths.

[7] We indicate embedding vectors for nodes $e$ and relations $r$ with bold symbols to contrast them to symbolic nodes $e$ and relations $r$ from the KG.

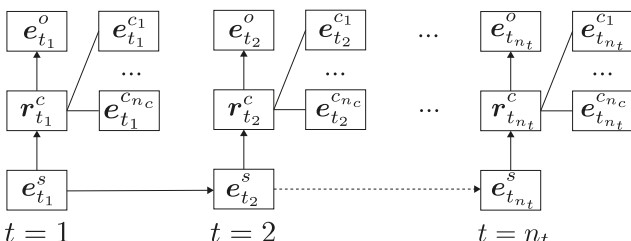

**Fig. 3.** Sequential context for subject and relation embedding $e^s$ and $r$. The object $e^o$ and contextual factors $e^c$ refer to a different symbolic KG entity $e^o$ and $e^c$ in every step. They are given by (pre-trained) static KGEs. In contrast $e^s$ and $r$ represent the same symbolic KG node $e^s$ and hyper-edge $r$, regardless of time and context. However, the embedding is customized with a situation-specific contextualized embedding, depending on the temporal and relational context.

## 5   RETRA: The Recurrent Transformer

Learning customized embeddings based on subjective sequential context requires a novel Neural Network (NN) architecture.

### 5.1   The RETRA architecture

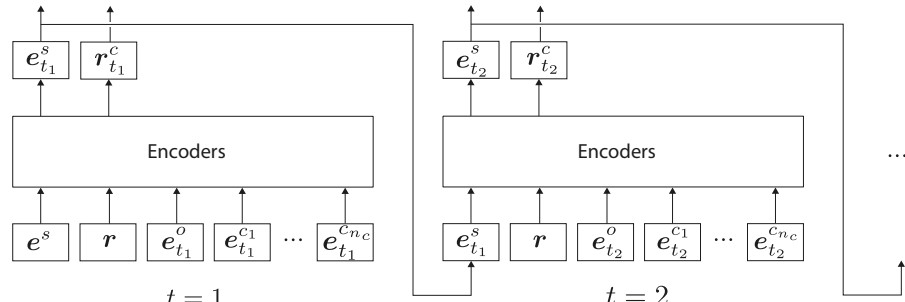

**Fig. 4.** Recurrency in the RETRA architecture: In the first step, $e^s$ is not temporally contextualized but a static KGE embedding. In $t = 2$ the contextualized $e^s_{t_1}$ is used as input to generate the temporally contextualized $e^s_{t_2}$

.

Our model is inspired by the encoder stack of transformers ([20]) and Recurrent Neural Networks (RNNs) and can thus be called a Recurrent Transformer (RETRA). We can't use common RNN architectures, like LSTMs [7], nor transformer models, since both don't handle multiple variable length inputs per step

in a temporal sequence. Regarding input and output, RETRA receives the pre-trained static embeddings $e^s, r, e^o, e^{c_1}, ..., e^{c_{n_c}}$ and outputs the contextualized $r^c$. In addition, $e^s$'s previous subjective memory $e^s_{t-1}$ is passed on to generate the temporally contextualized embedding $e^s_t$ for the current step. Thus, the only non-pre-trained embedding passed on to the next step is $e^s_{t-1}$ (cmp. Fig. 4).

The final crucial building block to transform $r \longrightarrow r^c$ and $e^s_{t-1} \longrightarrow e^s_t$ is handled inside the encoder stack. Similar to [20] we use a stack of encoder layers, each consisting of a self-attention layer followed by a feed forward network. We adapt each attention head in the self-attention layer to resemble the structure of the relations defined by a tcKG. Thus, we don't need to calculate the pairwise attention for all inputs to the encoder, but can attend $r^c$ only to $\{e^s, r, e^o, e^{c_1}, ..., e^{c_{n_c}}\}$. Similarly, we can constrain the attention of $e^s_t$ to $\{e^s_{t-1}, r^c_t, e^o_t\}$ only. This is displayed by the diagonal arrows inside the first encoder layer in Fig. 5.[8]

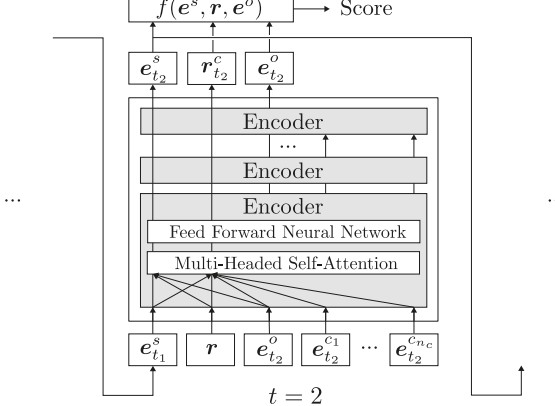

**Fig. 5.** Inside the Encoders in the RETRA architecture (cmp. Fig. 4): Stacked encoder-layers, each with constrained multi-headed self-attention, followed by the scoring function which returns a scalar score for $(e^s, r, e^o)$-triple.

## 5.2   Training RETRA

To optimize the weight matrices in the feed-forward and self-attention layers using backpropagation we need to measure the plausibility of predicted facts (using a scoring function) and its deviation from known facts given in the training data (using a loss function). In principal RETRA is independent of the choice of the scoring function and training objective (see Table 5 in [8] for an overview of

---

[8] The constraining inside the attention heads is an engineering choice and acts as an inductive bias. Any other constraining is possible including no constraining.

state-of-the-art KGE models and their scoring functions). Any scoring functions and training objectives can be plugged into RETRA as long as it allows to calculate a gradient.[9] We settle on the most common KGE training objective, namely link prediction, which we in "transformer terms" refer to as "object masking". The training target is to correctly predict $e_{t+1}^o$ given $e_t^s$ and $r_t^c$, where $e_{t+1}^o$ is masked out. Thus, the weights need to be adjusted, such that the scoring function $f(e_t^s, r_t^c, e_t^o)$ outputs a high score for the correct $e_{t+1}^o$ from the training data (and a low score for all other entities). Using a soft-max function on the predicted scores for each $e^o$s allows to calculate the cross-entropy loss against the correct triple and backprop the error.

In this paper we are only interested in optimizing the embeddings regardless of the scoring function provided. Thus, we compare the predictive performance of a given KGE model, including its scoring function, to using the same scoring function but transforming the embeddings to temporally contextualized KGEs.

## 6    Implementation and Empirical Testing

This section provides implementation details and reports empirical results from three diverse application domains. An overview of the features selected in each domain as tcKG facts is provided in Tab. 1. The SUMO dataset and the parameters used for training in the following experimental section are available in our repository[10] on Github.com. Once we get the internal approval, the code to run the experiments will also be made available there.

The proposed RETRA approach is implemented based on PyTorch's[11] Transformer Encoder layer, which provides an internal self-attention layer. As seen in Figure 5, we use an embedded triple $(e_{t_1}^s, r, e_{t_2}^o)$ plus its contexts $e^{c_1}, ..., e^{c_{n_c}}$ as input and assume the output to contain the contextualized embeddings $e_{t_2}^s$ and $r_{t_2}^c$. Of those embeddings, the contextualized subject embedding replaces or complements the global subject embedding in the next time-step. This is repeated over the whole sequence of experiences of $e^s$. By doing so, the subject embedding alters based on the history of previous inputs and its current context.

One key feature of RETRA is its complementarity to existing KGE methods. In the use-cases we present here, we use the static KGEs and their respective scoring functions from three established baseline KGE techniques, namely TransE [1], SimplE [10] and HolE [15]. The implementations of the baseline models were acquired using the OpenKE[12] framework, which offers fast implementations of various KGE approaches. The focus of this work is not to obtain the best overall predictive performance but to show how temporal contextualization can improve existing KGEs. For that reason we picked three basic and established baselines.

---

[9] Also, many other self-supervised training objectives are possible. Starting from relation masking to temporal subject masking (mask subject $e_t^s$ and condition the prediction on $e_{t-1}^s$).

[10] https://github.com/siwer/Retra

[11] https://pytorch.org/

[12] https://github.com/thunlp/OpenKE

### 6.1 Location Recommendation

For location recommendation, we use the New York City dataset[13] which was created and used for a different recommendation scenario before [23]. The data consists of check-ins from Foursquare[14], which is a location based social network. Every check-in consists of various information, including user, location, location type, country and time. The recommendation target is a *location* or *Point of Interest* (POI) for a particular user, given background knowledge about the locations and the history of *visits* or *checkIns* of users at POIs. The ranking is done by using a scoring function $f(e^s, r, e^o)$ provided by the baseline KGE approaches. The result of a forward step in this scenario is a tensor containing the scores for every potential location in the data. This information plus the information about the known target location given in the training data serves for calculating the cross-entropy loss.

**Location Recommendation - Experimental Design:** We consider the 'raw' setting provided in the data set, since we are treating every check-in as one distinct time-step. The data set contains 104,991 distinct check-ins, 3,626 distinct locations, 3,754 distinct users and 281 distinct types. For training and testing we were using a random 80 - 20 split of our data. Users with only one check-in were not considered because a sequence of at least two check-ins is needed for contextualization. In addition to the input of a triple $(e^s, r, e^o)$, we explicitly passed the preceding location and the current location's type as context. The check-ins are not uniformly distributed over users. There are many users with only one or two check-ins, and few users with a lot of check-ins (up to 4,069). The same pattern can be identified with the locations. This extreme imbalance makes this a very challenges task, since we assume that a longer sequence provides more information on a certain user's behaviour than a short sequence would do.

Basic KGE approaches are unable to incorporate the inherent sequential and $n$-ary relational information provided by such a dataset and are thus fundamentally limited for this task. For both approaches, we have chosen the default number of dimensions (130) as the embedding size.

**Location Recommendation - Experimental Results:** It can be seen in Table 2, that all baseline approaches have performance issues, which we attribute to the skewed distribution. Still, we use these approaches as our global baseline embeddings to see if it is possible to incorporate more information by modelling the sequence and the context information and thus obtain an increase in performance. When the baseline KGEs are combined with RETRA we indeed obtain a huge relative performance increase. Numbers in bold indicate the best results. While the overall performance is still low, the results show that the usage of sequential and contextual information for enhancing entity embeddings can improve the performance of standard KGE approaches by a factor of up to 15.

---

[13] https://github.com/eXascaleInfolab/LBSN2Vec
[14] https://foursquare.com

| Approach | Hit@10 | Hit@3 | Hit@1 | Imp Hit@10 | Imp Hit@3 | Imp Hit@1 |
|---|---|---|---|---|---|---|
| TransE | 0.0100 | 0.0017 | 0.0004 | - | - | - |
| SimplE | 0.0077 | 0.0035 | 0.0013 | - | - | - |
| HolE | 0.0038 | 0.0004 | 0.0 | - | - | - |
| RETRA+TransE | 0.0203 | 0.0005 | 0.0001 | 103% | -70% | -75% |
| RETRA+SimplE | **0.0592** | **0.0521** | **0.0194** | 668% | 1388% | 1392% |
| RETRA+HolE | 0.0209 | 0.0005 | 0.0 | 450% | 25% | 0% |

**Table 2.** Metrics for the best runs of the baseline and combined approaches. "Imp" refers to the relative percentage of performance change compared to the corresponding baseline metric.

When testing different combinations of model parameters, we observed that the learning rate has the strongest influence on the performance. Changing the number of transformer layers does not seem to have a big impact in general. Apparently, the interactions between features is not complex enough to require several attention layers. For all tested scoring functions, the combination with RETRA led to an improvement in performance.

### 6.2   Driving Situation Classification

Much progress has been made towards automated driving. One challenging task in automated driving is to capture relevant traffic participants and integrated prediction and planning of the next movement by considering the given context and possible interactive scenarios. Here, we define the problem as predicting the driving maneuver (e.g. following, merging, overtaking) of a vehicle given the current state of the driving scene. According to [12] approaches for vehicle motion prediction can be grouped into physics-based , maneuver-based and interaction-based . Interaction-based methods extend maneuver-based methods by modelling the dependencies between pairs of vehicles. Related work based on different deep neural network approaches and feature combinations for trajectory prediction has been described in [13] in which surrounding vehicles and their features are extracted from fixed grid cells. Our approach in comparison uses relational data between the ego and foe vehicles. Our motivation is that explicit representation of triples might lead to improved modelling of interactions between vehicles.

**Driving Situation Classification - Experimental Design:** We use SUMO[15] (Simulations of Urban Mobility), an open source, highly portable, microscopic and continuous multi-modal traffic simulation package to generate driving data. More than 50′000 driving scenes of a motorway were generated. The vehicle parameters as well as driving styles were varied widely in order to simulate a large

---

[15] https://www.eclipse.org/sumo/

variety of vehicles and driving behaviours. This resulted in situations such as risky driving situations, abandoned driving maneuvers, unexpected stops and even accidents. We have developed a knowledge graph to represent the simulated data by entities (e.g. scene, situation, vehicle, scenario), relations between entities (e.g. isPartOf, occursIn, type) and their associated features (e.g. speed, acceleration, driving direction, time-to-collision). This resulted in more than 900 Mio. RDF-triples with around 2 million scenes which comprise more than 5 million *Lane Change* and *Conflict* situations, respectively. It represents a valuable benchmark data-set for driving situation analysis. More information on the design and creation process of the data-set is available in [6].

| Approach | Sequence Length | Hit@3 | Hit@1 | MR | MRR |
|---|---|---|---|---|---|
| HolE | 0 | 0.9366 | 0.5235 | 1.76 | 0.72 |
| TransE | 0 | 0.7668 | 0.2729 | 2.56 | 0.53 |
| SimplE | 0 | - | - | - | - |
| RETRA+FF | 0 | **0.9946** | 0.8060 | 1.23 | 0.89 |
| RETRA+FF | 5 | 0.9731 | 0.8212 | 1.23 | 0.90 |
| RETRA+FF | 10 | 0.9672 | 0.8382 | 1.17 | 0.91 |
| RETRA+FF | 15 | 0.9858 | 0.8455 | 1.17 | **0.92** |
| RETRA+FF | 20 | 0.9871 | **0.8469** | **1.16** | **0.92** |

**Table 3.** Results for the SUMO Driving Situation Classification data set. The task was to predict the correct situation type, given surrounding traffic. All performance metrics (hit@k, mean rank (MR) and mean reciprocal rank (MRR)) indicate that context is crucial and more previous observations information improves the performance more.

**Driving Situation Classification - Experimental Results:** We conducted two sets of experiments on the SUMO data, which both aimed at predicting the type of a conflict. We needed to make this distinction since the baseline KGE methods cannot use context and thus have to make predictions based on the situation-ID. Instead, RETRA learns a dedicated situation embedding based on the context and previous driving scenes. When experimenting with the different baseline KGE scoring functions we noticed that a fully connected feed forward layer (FF) as a trainable scoring function performs better. The results are shown in the first four rows of Table 3. Since various SimplE implementation we tried did not scale to the size of this data set, we can't report any results. Obviously, RETRA+FF considerably outperforms the baselines, even as a non-recurrent version. This is mostly due to its ability to contextualize a situation embedding which avoids the need for explicit situation-IDs.

Since the previous steps in time leading up to the current situation are potentially important in driving scenes, we specifically investigated the influence of

previous situations on the predictive performance. The last four rows of Table 3 show how RETRA handles different numbers of recurrence steps, by feeding in the preceding 5 - 20 driving situations leading up to the current point in time. It can be observed that the longer sequences are the better the results get. This confirms the assumption that the history is important in driving situations and RETRA is able to exploit it.

### 6.3   Event Prediction

The *Integrated Crisis Early Warning System* [2] contains information on geopolitical events and conflicts and is a widely used benchmark for both static and temporal KGE approaches. We specifically use this dataset to showcase how contextualizing can improve and generalize binary KGE approaches.

**Event Prediction - Data Set:** For our experiments, we use the 2014 subset[16] of the ICEWS data as described in [5] as a basis, and add contextual information that we take from the original 2014 data[17]. In addition to the triples consisting of *Source*, *Event Text* and *Target*, we use the entities *Source Sector*, *Source Country*, *Target Country* and *Intensity* to contextualize the *Source*.

**Event Prediction - Experimental Results:** The target is to predict the *target entity*, typically the organization involved in the event, given the *source entity*, aka actor, and the relation. In both setups, we optimize a cross-entropy loss by calculating scores for all possible triples in a query $(s, r, ?)$. The target is to produce the highest score for the original triple given in the ground-truth. In addition to using only the information presented in triples, we also consider contextual information for our training. This is achieved by passing all information through RETRA and using the contextualized subject entity for the query $(s_c, r, ?)$. In this way, the embeddings are learnt in such a manner that they contribute to the contextualizing given a binary scoring function from our baseline KGE methods. As shown in Table 4, using the contextual information results in a huge improvement in performance for all tested baseline scoring functions and evaluation metrics. This, again, indicates that context is crucial and RETRA is able to exploit it, regardless of the KGE scoring function used.

## 7   Conclusion and Future Work

In this paper we propose the modeling template **tcKG** for temporally contextualized KG facts (addressing *RQ1*) and **RETRA**, a Deep Learning model intended to transform static Knowledge Graph Embeddings into temporally contextualized ones, given a sequence of tcKG facts (addressing *RQ2*). With RETRA we tackle two limitations of current KGE models, namely their lack of taking *n*-ary

---

[16] https://github.com/nle-ml/mmkb/tree/master/TemporalKGs
[17] https://dataverse.harvard.edu/dataset.xhtml?persistentId=doi:10.7910/DVN/28075

| Metric | Contextualized | | | Non-Contextualized | | |
|---|---|---|---|---|---|---|
| | TransE | SimplE | HolE | TransE | SimplE | HolE |
| Hits@1 | 0.519 | **0.570** | 0.537 | 0.264 | 0.264 | 0.299 |
| Hits@3 | 0.691 | **0.739** | 0.703 | 0.398 | 0.398 | 0.463 |
| Hits@10 | 0.821 | **0.843** | 0.822 | 0.532 | 0.532 | 0.623 |
| Hits@100 | **0.941** | **0.941** | 0.940 | 0.775 | 0.775 | 0.840 |
| MR | 152.92 | 91.02 | **88.00** | 311.85 | 311.85 | 193.13 |
| MRR | 0.625 | **0.669** | 0.638 | 0.358 | 0.358 | 0.409 |

**Table 4.** Contextualized vs. Non-contextualized KGE for different scoring functions on the ICEWS event prediction data set.

relational context into account (*Lim1*) and capturing the evolution of an entity embedding, given its subjective history of similar previous events (*Lim2*). Our experimental results on three data sets from diverse application domains indicate that existing KGE methods for global embeddings can benefit from using RETRA to contextualize their embeddings. We could also demonstrate that both, context and history, does boost performance considerably.

Although there have been a number of recent contributions to the area of contextualized KGEs, we still see large potential for future work beyond additional empirical testing and technical improvements to RETRA. From the perspective of knowledge representation the fundamental question remains how to best capture influencing factors that contextualize the meaning of an entity or relation. We see this as a crucial challenge for making KGs more actionable in concrete real-world situation. Once this is solved efficiently, we expect a similar boost to KGEs as Transformers generated for word embeddings.

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
