# OpenReview forum: "RETRA: Recurrent Transformers for Learning Temporally Contextualized Knowledge Graph Embeddings"
_eswc-conferences.org/ESWC/2021/Conference/Research_Track — ESWC 2021 Research_

### Official Review · AnonReviewer2 · 2021-01-04
**A lot of formalization for unclear application**

**Rating:** 1
**Confidence:** 2
**Impact:** 4
**Design And Technical Quality:** 3

**Review:**

For this paper, the motication and application is very unclear. The authors introduce their paper with the sentence "We all play different roles in our lives." people might act differently in different contexts, but for the process of identification, we are still the same human with the same characteristics. For the motivation of this approach, the paper clearly lacks a proper example. I am not able to comprehend what the authors intend to represent.
In knowledge graphs, it is essential to identify the entities uniquely. Each entity is a node in the graph with relationships to other nodes.
What do the authors mean with "concreate situations"? Is it the fact, that territorials of countries changed over time and cities were renamed or did play a different role in different times? Or do they want to represent entities in concrete roles, such as a person who is an actor for one movie and a director for another one?
For their formal graph representation, graph edges are supplemented with time(stamps). In this way, the graph is evolving along a timeline.
The actual contribution is then described ins section 5: a graph transcoder. The authors introduce the architecture as well as the training process. They state that RETRA is able to handle different scoring functions and reference a table in another paper. Here, a short description of this referenced overview would have been helpful.
RETRA is evaluated utilizing three different application domains: lov´cation recommendation, driving situation classification and event prediction. At this point, possible applications for RETRA are described for the first time.
The authors compare RETRA to three baseline techniques: TransE, SimplE and HolE. All of the have been acquired using OpenKE. as the authors state, their intention is to show the impact of temporal contextualization on existing KGE.
For two of the three evaluation settings, a contextualized SimplE seems to outperform the other scoring functions. For the drinving situation classification none of the baselines benefitted from the cintextualization. Instead, a fourth scoring function - the fully conntect feed forward layer - outperforms the other baselines. It would be helpful to discuss this aspect a bit more in detail.
Overall, the authors present a promising approach using contextualization KG embeddings.
The paper lacks a clear motivation and a leading example to fully comprehend what the authors are aiming at.

---- after rebuttal ----

Thanks to the authors for the comments. Although, I still think the paper could benefit from some examples to clarify the application, I would like to see the approach presented at the conference.

**Anonymity:**

Yes, I would like my review to remain anonymous.

**Reuse And Availability:**

3: Medium

**Strong Points:**

- thoughtful evaluation on three different domains utiling baseline scoring functions
- elaborate formal description of temporal context

**Subreviewer:**

I submitted this review.

**Weak Points:**

- motivation and application is not clear when describing the approach - application domains are firstly introduced in the evaluation section

---

> ### Author Rebuttal · Authors · 2021-01-29
>
> Thanks for your inspiring feedback, and opening up a very broad discussion on modelling roles in knowledge graphs.
>
> To clarify one issue right away, we still rely on uniquely identifying entities in the graph. We are not attempting to introduce the same entity several times, depending on the context. We are proposing a modelling pattern that allows to identify the relevant context for an entity, depending on the role of the entity in a concrete situation.
>
> We ourselves are not yet sure how general our tcKG patterns are and where they can be applied. They certainly are flexible as we showed in our three very heterogenous application scenarios. That is also the reason why we chose not to focus on a single leading example. We wanted to emphasize the generality of tcKG patterns.
>
> Your example of a “person who is an actor for one movie and a director for another one” certainly is another intuitive example. Our key assumption is that in every-day situation a person is making decisions with a subjective focus on selected properties and experiences only. A prediction about how a person is rating a movie might be very different if he is the director or an actor (or not involved in the movie). This focus might be defined by his role, e.g., actor vs. director. Traditionally, KGs are not intended to capture such information, since they focus on objective and universal facts about the entity. However, we argue that temporal context is essential for decision making in complex scenarios and should be paid more attention in KG research in order to extend the applicability of KG to other types of tasks.
>
> Concerning scoring functions, we want to present RETRA as a general formalisation framework, which works for all kinds of different scoring functions. If, for instance, a well-established scoring function doesn’t perform well on data with a certain (for example temporal) structure, RETRA allows to boost its performance by allowing it to exploit temporal information as well. Regarding the issue with the FF network as scoring function in the driving scene classification, our motivation was to show the flexibility of our approach. The overall formalization stays the same, while the approach can be easily adapted to new data by replacing just the scoring function.

---

### Official Review · AnonReviewer3 · 2021-01-15
**Interesting approach and evaluation with limited applicability and weak comparison**

**Rating:** 1
**Confidence:** 2
**Impact:** 3
**Design And Technical Quality:** 4

**Review:**

This submission proposes a more comprehensive approach for knowledge graph embeddings (KGE). It focuses on two aspects: 1) modeling of contextualized information; 2) modeling of temporal sequences. Recent methods for KGE have been proposed that tackle aspect 1 or 2, but not both. The approach is described in much detail. It is evaluated on three tasks: location recommendation, driving scene prediction, and event prediction, showing improvements over traditional KGE methods.

Strong points:

S1. the contribution of the paper is clearly described

S2. the paper is well-written and well-structured

S3. the method is clear and described in detail

S4. the tasks are interesting/creative and relevant for the method

S5. there are some interesting findings in the evaluation section

Weak points:

W1. The set of baselines is quite limited. State-of-the-art contextualized baselines: DOLORES, StarE, KG-BERT, HINGE; or temporal KGE baselines should be included. It is unclear why the paper does not include any of these, as they would allow for much more meaningful evaluation.

W2. The results for experiment 1 are quite low, with no commentary. Also, TransE is outperforming RETRA on the lower HITS metrics - some commentary or further analysis would be welcome here.

W3. While it is exciting to have a method that works for both temporal sequences and n-ary relations (in hypergraphs), it is not clear how useful this method is. I appreciate the creativity in choosing the evaluation tasks, which mostly rely on both the contextual and the temporal aspect; but I am wondering if this method is relevant for more fundamental/established tasks too.

W4. I miss a discussion on the upshot of the results - what have we learned about this novel method? What worked and what did not?

Other comments:
* please be specific about the task definition  and the ground truth in the location recommendation task
* In table 4, please comment on why TransE and SimplE have identical results for the non-contextualized columns


---- After rebuttal ----

I thank the authors for the rebuttal. While all of my questions have been addressed, I feel like none of them has been really answered. For example, in W3 I asked for examples of other applications, and the authors reply that "in the real world almost all tasks depend on context and time".

**Anonymity:**

Yes, I would like my review to remain anonymous.

**Reuse And Availability:**

3: Medium

**Subreviewer:**

I submitted this review.

---

> ### Author Rebuttal · Authors · 2021-01-29
>
> Thanks for your constructive feedback, we hope that we can clarify the issues you raised:
>
> W1: The reason for the limited set of baseline KGE techniques is due to the fact that RETRA is a general framework. RETRA is not meant to replace KGEs but to extend them towards temporal AND contextualized embeddings. Thus, our intention is to show the improvement of an existing KGE technique when implemented inside the RETRA framework. We expect that the obtained gains are similar for any KGE approach, including the temporal OR contextualized ones mentioned in your review. In the end, the baselines used in our paper were chosen for their simplicity and ease of integration of their scoring functions with RETRA.
>
>
> W2: In experiment 1, the issues with the low metrics are due to the dataset structure. Triple-based scoring functions without context apparently do not seem to work well on this dataset, likely due to the skewed distribution. This is empirically evident but the “why” was hard to justify.
>
> W3: Thank you for this very interesting question: In our opinion this approach is broadly applicable, since it is an extension of and not a replacement for established tasks. For instance, link prediction can be extended to temporally contextualized link prediction. With RETRA, context and temporal dependencies can be modelled and exploited. In the real world, almost all tasks depend on context and time. However, in practice such information is often not recorded and the standard benchmarks are missing this type of information (see also the reply to AnonReviewer2).
>
> W4: The benefit of our method is that it enables generalizing well established approaches to a wider range of applications. Our results show that – dependent on the data used – we can improve predictive results by adding contextual and/or temporal information into the training process, without the need of defining a completely new approach, but instead by generalizing established methods.
>
> Regarding Table 4, we noticed that we reported some results incorrectly. The corrected version for the Non-Contextualized SimplE runs is as follows: Hits@1 = 0.316, Hits@3 = 0.4817, Hits@10 = 0.6433, Hits@100 = 0.8576, MR = 183.59, MRR = 0.426. We apologize, and will correct this in the final version of the paper.

---

### Official Review · AnonReviewer4 · 2021-01-15
**Review of RETRA**

**Rating:** 3
**Confidence:** 5
**Impact:** 4
**Design And Technical Quality:** 5

**Review:**

This is a very good piece of work, well motivated (with identified limitations of existing approaches), well prepared, well well illustrated, well evaluated & well written.

Minor details:
- check the manuscript for possible typos, I only spotted “embedd” in 2.3 but there can be more

**Anonymity:**

No, I would like my review to be deanonymized.

**Reuse And Availability:**

5: Very High

**Strong Points:**

I really like how the authors motivate the setting for the paper, and both argue for the need for tcKG and RETRA, and support the arguments by careful analysis of limitations of existing systems.

**Subreviewer:**

I submitted this review.

**Weak Points:**


Although I fully enjoyed reading the paper it made me think that it would be nice to read even more of the arguments, and especially see a few examples listed in table 1 also as graphs. That said, it is not really a weak point of the paper, just a matter of taste of representing examples.

---

> ### Author Rebuttal · Authors · 2021-01-29
>
> Thank you for your overall positive assessment and appreciation of our work.

---

### Official Review · AnonReviewer1 · 2021-01-19
**Interesting work**

**Rating:** 1
**Confidence:** 4
**Impact:** 3
**Design And Technical Quality:** 4

**Review:**

This paper motivates and presents an approach for incorporating temporally contextualized info into knowledge graph embeddings and develops a recurrent architecture with transformers for learning from these temporally contextualized knowledge graph embeddings. The model with contextualized KGEs gains performance improvements compared to the one without them in the empirical evaluation against three context-past-based predicting use cases including location recommendation, event prediction, and driving situation.

- The paper is technically sound, well-explained, and easy to follow.
- Incorporating contextual info with n-ary relationships into knowledge graph embeddings is interesting and useful for many learning and predicting tasks. This work can also motivate other problems related to contextualized knowledge graphs as well.
- The combination of the contextualized KGEs and the recurrent transformer model has gained good performance improvements compared to the non-contextualized versions.
- Minor grammatical errors should be corrected

Questions:
- How long does it take for each of the training and predicting tasks?
- What are the training parameters and their ranges? Batch size, steps, embedding size, input size, learning rate, etc?
- How can overfitting be addressed in the model training?

**Anonymity:**

Yes, I would like my review to remain anonymous.

**Reuse And Availability:**

3: Medium

**Strong Points:**

- The approach is technically sound and well-explained in general
- Performance gain is significant compared to the one without contextualized info
- Adding contexts into knowledge graph embeddings is an important task and it's well demonstrated

**Subreviewer:**

I submitted this review.

**Weak Points:**

- The evaluation section should discuss the training parameters in more detail and how they affect the performance. These are currently missing and it does not shed much light on the training process.

---

> ### Author Rebuttal · Authors · 2021-01-29
>
> Thank you for your constructive feedback, and your questions towards training parameters.
>
> The exact experimental setup will be contained in the repository with the code and data once the paper will be published. Some answers to this question are provided below:
>
> The duration of training + predicting in our experiments varies based on the given conditions and parameters. The reported experimental results were achieved using the parameters mentioned below.
>
> Overfitting was addressed by applying dropout (=0.2) in the encoder layer and by adjusting the parameters (reducing the embedding dimension, forward dimension and nr of attention heads and stacked encoder layers).
>
> All training/evaluation/testing was executed on GeForce RTX 2080 Ti.
>
> - SUMO:
>   - Adam optimizer, learning rate = 0.00005
>   - Batchsize = 2048, trained for 100 epochs, embedding dimension = 50, encoder forward dimension = 10
>   - 5 encoder layers stacked using 5 attention heads
>   - Training + Evaluation time: between 5 to 10 minutes
>
> - ICEWS:
>   - Adagrad optimizer, no decay, learning rate = 0.008
>   - Batchsize = 256, trained for 15 epochs, embedding dimension = 50, encoder forward dimension = 50
>   - 1 encoder layer and 1 attention head
>   - Training + Evaluation time: around 15 minutes
>
> - FOURSQUARE:
>   - Adagrad optimizer, no decay, learning rate = 0.09
>   - Batchsize = 1, trained for 10 epochs, embedding dimension = 130, encoder forward dimension = 130
>   - 1 encoder layer 1 attention head
>   - Training + Evaluation time: around 12 – 14 hours*
>
> *the reason for this is the batch size of 1, which is a consequence of the skewed distribution (many users with 1 to 2 check-ins, few users with check-ins > 20). Using variable-sized batches does not provide a substantial speed-up, because it is the very long sequences (> 100 check-ins) that seem to slow down training). Alternatively, more data preprocessing (like restricting the sequence lengths to a certain size) would have been needed. We decided against this to keep our approach comparable to the baseline approaches.

---

### Decision · Program_Chairs · 2021-02-23

**Decision:**

Accept

**Comment:**

This paper motivates and presents an approach for incorporating temporally contextualized information into knowledge graph embeddings and develops a recurrent architecture with transformers for learning from these temporally contextualized knowledge graph embeddings. The paper is interesting, relevant for the ESWC community and provides mature contributions. As reviews are positive, we recommend acceptance.


To address the comments of reviewers, the authors are asked to expand the evaluation section and extend the discussion regarding the applications of the introduced method.